# Shifting the Cancer Screening Paradigm: The Rising Potential of Blood-Based Multi-Cancer Early Detection Tests

**DOI:** 10.3390/cells12060935

**Published:** 2023-03-18

**Authors:** Tiago Brito-Rocha, Vera Constâncio, Rui Henrique, Carmen Jerónimo

**Affiliations:** 1Cancer Biology and Epigenetics Group, Research Center (CI-IPOP)/RISE@CI-IPOP (Health Research Network), Portuguese Oncology Institute of Porto (IPO-Porto)/Porto Comprehensive Cancer Center Raquel Seruca (P.CCC), Rua Dr. António Bernardino de Almeida, 4200-072 Porto, Portugal; 2Master Program in Oncology, School of Medicine & Biomedical Sciences, University of Porto (ICBAS-UP), Rua Jorge Viterbo Ferreira 228, 4050-513 Porto, Portugal; 3Doctoral Program in Biomedical Sciences, School of Medicine & Biomedical Sciences, University of Porto (ICBAS-UP), Rua Jorge Viterbo Ferreira 228, 4050-513 Porto, Portugal; 4Department of Pathology, Portuguese Oncology Institute of Porto (IPO-Porto), Rua Dr. António Bernardino de Almeida, 4200-072 Porto, Portugal; 5Department of Pathology and Molecular Immunology, School of Medicine & Biomedical Sciences, University of Porto (ICBAS-UP), Rua Jorge Viterbo Ferreira 228, 4050-513 Porto, Portugal

**Keywords:** cancer screening, multi-cancer early detection, MCED, liquid biopsy, biomarkers

## Abstract

Cancer remains a leading cause of death worldwide, partly owing to late detection which entails limited and often ineffective therapeutic options. Most cancers lack validated screening procedures, and the ones available disclose several drawbacks, leading to low patient compliance and unnecessary workups, adding up the costs to healthcare systems. Hence, there is a great need for innovative, accurate, and minimally invasive tools for early cancer detection. In recent years, multi-cancer early detection (MCED) tests emerged as a promising screening tool, combining molecular analysis of tumor-related markers present in body fluids with artificial intelligence to simultaneously detect a variety of cancers and further discriminate the underlying cancer type. Herein, we aim to provide a highlight of the variety of strategies currently under development concerning MCED, as well as the major factors which are preventing clinical implementation. Although MCED tests depict great potential for clinical application, large-scale clinical validation studies are still lacking.

## 1. Introduction

Cancer represents a major public health concern, being the leading cause of death in most countries. Indeed, 10 million deaths and 19.3 million new cancer cases were estimated worldwide in 2020 [1]. This high mortality rate is mostly due to late detection, finding cancer when it has already progressed and metastasized, which significantly reduces effective treatment options. It is estimated that at least 15% of cancer-related deaths within 5 years could be avoided by early disease detection [2]. Hence, cancer screening and early detection should be prioritized, preventing cancer development by removing pre-cancerous lesions and avoiding its progression by effective treatment of localized disease [3,4]. Nonetheless, only a handful of cancer types have recommended screening procedures. The United States Preventive Services Task Force (USPSTF) recommends population-based screening for lung (in high-risk individuals), colorectal, breast, and cervical cancer, while in European countries, only the latter three tumor types have approved screening programs [5,6]. In addition, prostate cancer screening is available in the US, although on an individual basis [7]. Thus, more than 60% of cancer-related deaths are caused by malignancies for which there is no screening test available [1].

Although the adoption of screening programs has indeed contributed to increased survival rates in those cancer types, many factors are hampering screening from reaching higher level efficacy. For instance, lung and breast cancers are detected by low-dose CT and mammography, respectively, which, besides exposing individuals to radiation, eventually lead to some overdiagnosis and false positive results [8,9]. The same applies to cervical and prostate cancer screening, based on cytology/HPV and serum PSA testing, respectively. [10,11]. Contrarily, colonoscopy allows for a very accurate detection of colorectal cancer, as well as its precursor lesions and their subsequent removal. However, it is a rather invasive and uncomfortable procedure, requiring prior preparation, which results in low patient compliance [12] (Figure 1).

At present, following an abnormal finding in a screening procedure, a tissue biopsy must be conducted for histopathological evaluation and eventual cancer diagnosis. In fact, tissue sampling has been the gold standard approach for cancer diagnosis and prognostication, but several disadvantages can be pointed out to the use of this biological material: (1) it requires an invasive collection procedure; (2) some tumors are not easily accessible due to their anatomical location; (3) it has limited ability to be used as an early detection tool; (4) it has limitations in the evaluation of treatment efficacy and monitoring of tumor progression; and (5) it does not fully represent tumor heterogeneity [13,14]. Thus, minimally invasive techniques allowing for improved disease detection and monitoring are desirable. Recently, liquid biopsies have emerged as tools to overcome these challenges. Consisting in the analysis of disease-related markers from body fluids, such as blood or urine, liquid biopsies comprise a variety of analytes, namely, circulating cell-free DNA (cfDNA), cell-free RNA (cfRNA), circulating tumor cells (CTCs), extracellular vesicles (EVs), tumor-educated platelets (TEPs), proteins and metabolites [15,16]. The analysis of these biomarkers enables the identification of tumor-related information and, consequently, tumor burden real-time monitoring, thereby having great potential to improve routine clinical practice [17]. Furthermore, because tumors shed these analytes into the circulation early in their development, liquid biopsies have the capacity to detect cancer even when symptoms are not present or tumor masses are not detectable by imaging techniques [18,19]. Considering the hurdles faced by current cancer screening paradigms, a blood-based test that might simultaneously detect multiple cancer types, at early stages, and even be applied to high-risk population-based screening, constitutes an exciting and clinically valuable tool. Moreover, a pan-cancer approach might be the only cost-effective option for screening of low prevalent cancers [20]. Ideally, such a multi-cancer early detection (MCED) test should have high sensitivity for early-stage disease detection, high specificity to avoid false-positive results, and the ability to discriminate the tissue of origin (TOO) of the detected cancer [20].

Having this in mind, we conducted a literature review aiming to explore the diversity of strategies currently under development for multi-cancer early detection. Thus, a PubMed search was performed with the query (pan-cancer OR multi-cancer) AND (detection OR screening OR diagnosis) with no time interval restrictions. In total, 675 results were retrieved and imported to Rayyan, an intuitive website for title and abstract screening [21]. Additionally, 42 articles found from other sources were included. All abstracts were critically evaluated to select only those providing relevant information related to the topic of interest. Furthermore, only articles written in English, presenting original data and reporting biomarker performance metrics (AUC, sensitivity, specificity, etc.) were considered. A summary of the methodology is shown in Figure 2. The information gathered from the included studies is displayed in Table 1 and Table 2, showing multi-cancer detection strategies validated in human clinical specimens or based on data mining, respectively. Finally, a search was conducted on the ClinicalTrials.gov webpage to look for relevant clinical studies evaluating MCED tests, and the respective results are shown in Table 3.

## 2. Multi-Cancer Early Detection (MCED) Tests: State of the Art

### 2.1. Mutation-Based MCED Tests

Molecular profiling of driver mutations in tumor tissue has been the main strategy to assess cancer prognosis, treatment response monitoring, and resistance detection, as well as to detect disease recurrence. Accordingly, the current major clinical application of liquid biopsies is the detection of these mutations in tumor-derived cfDNA, i.e., circulating tumor DNA (ctDNA), to replace multiple puncturing with multiple blood draws [22,23]. Not surprisingly, MCED strategies have also relied on the detection of tumor-specific genetic variants in body fluids. As early as 2009, Zou et al. performed targeted mutation analysis in stool from several gastrointestinal cancer patients and showed that pan-gastrointestinal cancer detection was feasible with 68% sensitivity and 100% specificity [24]. In fact, stool is also a non-invasive source of cancer biomarkers but mostly limited to tumors of the digestive system. Interestingly, a study evaluating patients’ perceptions about stool-based multi-cancer detection reported that 98% of participants would use such a test, preferring it over conventional colorectal cancer screening, and highlighted its pan-cancer feature as the most relevant [25]. Subsequently, Quantgene Inc. developed DEEPGEN^TM^, a blood test based on next-generation sequencing (NGS) that detects low-frequency genetic abnormalities at a variant allele frequency of 0.09% [26,27]. When applied to the detection of seven cancer types, this assay displayed 43% sensitivity at 99% specificity with an area under ROC curve (AUC) of 0.90. Remarkably, an AUC of 0.88 was obtained for stage I cancer detection [28]. Cohen et al. also reported another blood test, CancerSEEK, for detecting eight common cancers (lung, breast, colorectal, pancreatic, gastric, hepatic, esophageal, and ovarian) based on the analysis of mutations in 16 genes combined with the circulating levels of eight proteins. Methodologically, this test consists of a multiplex PCR and a single immunoassay, constituting a simple workflow, easily applicable to clinical practice, with an estimated price of around USD 500. When applied to 1005 cancer patients and 812 healthy controls, CancerSEEK disclosed 62% sensitivity at a specificity greater than 99% for discriminating cancer from healthy samples. Concerning early-stage detection, a median sensitivity of 43% was observed for stage I, 73% for stage II, and 78% for stage III. Additionally, TOO discrimination was accomplished with 63% accuracy [29]. However, it is noteworthy that protein biomarkers were the major contributors to cancer type identification following a positive test result. A refined version of CancerSEEK was then developed in combination with PET-CT imaging to evaluate the test performance for prospectively detecting cancer in a study (DETECT-A) involving 10,006 women not known to harbor cancer. For that purpose, participants were blood tested, and, if abnormal, a second blood collection was conducted for confirmation and, if confirmed positive, a full body PET-CT was performed. Test results were considered positive for 134 participants, out of which 127 were further evaluated by PET. Sixty-four depicted suspicious imaging findings and 26 were proven to have cancer. This resulted in 27.1% sensitivity and 98.9% specificity for blood testing alone, while sensitivity decreased to 15.6% and specificity increased to 99.6% for blood testing combined with PET-CT imaging [30].

Therefore, although mutation-based MCED tests have demonstrated great capacity for cancer detection, even in early stages, these might not be the ideal standalone approach, since accurate TOO identification is difficult, due to a lack of tissue-specific gene driver mutations [31]. In fact, TOO discrimination is an essential feature of a MCED test, otherwise, individuals with a positive test would have to undergo additional costly exams for full body examination, instead of a confirmatory localized search [32,33]. Contrarily, epigenetic signatures are unique to each differentiated cell type, regulating its gene expression profile, thereby constituting a cell- and tissue-specific trait [34]. Indeed, DNA methylation patterns have demonstrated the capacity to distinguish tumor types in tissue samples [35] and also body fluids [36,37], as cfDNA fragments carry the methylation patterns of their cell of origin.

### 2.2. DNA Methylation-Based MCED Tests

DNA methylation, the most well studied epigenetic mechanism, consists in the addition of a methyl group to the 5-carbon of cytosines within CpG dinucleotides. While most CpG dinucleotides are scattered across gene coding regions and repetitive sequences, CpG clusters can be found in the so-called CpG islands, which are mostly present in gene promoters and first exons. In normal cells, CpG islands tend to be unmethylated, while coding and repetitive sequences are methylated. However, this methylation pattern is reversed in cancer cells, with promoters becoming hypermethylated, leading to tumor suppressor genes silencing, along with global hypomethylation, entailing genomic instability [38,39,40]. This aberrant methylation is thought to occur very early in the carcinogenic process, rendering DNA methylation an attractive biomarker for early cancer detection, alongside its TOO discrimination capacity and easy access through liquid biopsies [41]. Remarkably, about 50% of the studies selected for this review (Table 1) used DNA methylation as their approach for MCED.

Whether analyzing a single gene [42] or gene panels [43,44], cfDNA methylation levels have demonstrated the feasibility of using minimally invasive procedures to detect multiple cancers and further identify their anatomical location. Nonetheless, these approaches fall short regarding sensitivity values. Moreover, sequencing-based methylation profiling of cfDNA has shown more promising results, through the use of machine learning algorithms that convert the complex data acquired into classifiers that discriminate cancer from healthy individuals and further identify its origin. For instance, Kandimalla et al. reported EpiPanGI Dx, an assay that simultaneously detected gastrointestinal cancers with an AUC of 0.88 and 85–95% accuracy for TOO prediction [45]. Focusing on four major cancers (lung, breast, colorectal, and liver), the IvyGeneCORE^®^ Test developed by the Laboratory for Advanced Medicine demonstrated that methylation analysis of target genes discovered by data mining could detect these cancers with 84% sensitivity and 90% specificity [46,47]. Similarly, the PanSeer assay developed by Singlera Genomics [48] uses semi-targeted PCR libraries followed by sequencing for analyzing 477 differentially methylated regions (DMRs). This blood test was evaluated using samples from the Taizhou Longitudinal Study, in which healthy individuals provided plasma samples and were monitored for cancer development, allowing for a retrospective take on early detection viability. Concerning five tumor types (lung, colorectal, gastric, liver, and esophageal), 87.6% sensitivity and 96.1% specificity were observed, with similar sensitivity between early- and late-stage disease. Remarkably, using pre-diagnostic samples, PanSeer showed that cancer may be detected up to 4 years before medical diagnosis with 95.7% sensitivity [49]. Nevertheless, no results regarding TOO prediction were reported. At the time of writing, a clinical trial sponsored by Singlera Genomics (NCT05159544) was recruiting for a prospective study aiming to evaluate a multi-omics blood test for pan-cancer screening (Table 3).

A company that revolutionized the cancer screening paradigm and emphasized the wide variety of cancers that can be simultaneously detected through liquid biopsy is GRAIL, a spin-off of Illumina, that received around USD 1 billion in funding for the sole goal of developing a blood test for early cancer detection [50,51]. For such purpose, the Circulating Cell-free Genome Atlas Study (CCGA) (NCT02889978), divided into three sub-studies, was conducted and recruited over 15000 participants with and without cancer that were longitudinally followed up. In the first CCGA sub-study, three different sequencing assays were evaluated and, ultimately, whole-genome bisulfite sequencing outperformed whole-genome sequencing and targeted mutation analysis, demonstrating, once more, the superiority of DNA methylation analysis for early cancer detection [52,53]. Therefore, in the second sub-study, a targeted methylation assay was developed, trained, and validated using 6689 participants, for simultaneous detection and TOO discrimination of more than 50 cancer types. In this study, 54.9% sensitivity and 99.3% specificity were disclosed for all cancer stages, whereas 43.9% sensitivity was observed in early stages. Furthermore, when focusing on a set of 12 high-signal cancers (based on Surveillance, Epidemiology, and End Results (SEER) mortality data) sensitivity was 67.3%. Notably, 93% accuracy was displayed for TOO localization [54]. In the third and final sub-study, carried out to further validate an improved test version specific for screening purposes, an independent validation set of 5309 participants was used and resulted in 51.5% sensitivity, 99.5% specificity, and 88.7% accuracy for TOO prediction [55]. Considering the prospective nature of CCGA, the prognostic value of this blood test was also assessed. By following-up cancer patients from the second sub-study for 3 years, it was observed that cancers not detected by the test had significantly better overall survival (OS) than those detected by the MCED test. Additionally, detection sensitivity was higher for participants who died than in those who were alive, indicating that this test may improve the detection of aggressive cancers, thus being less prone to overdiagnosis [56]. Currently, this blood test is commercially available as Galleri^®^ at the price of USD 949, upon request to health care providers [57]. In addition to CCGA, other clinical trials are being conducted by GRAIL to ripen the tests’ potential as a screening tool (Table 3): STRIVE (NCT03085888) is evaluating the test performance to detect breast and other invasive cancers in women undergoing screening mammography; SUMMIT (NCT03934866) is evaluating the test performance to detect invasive cancers in individuals at high risk of lung and other cancers due to a significant smoking history; PATHFINDER (NCT04241796, NCT05155605) is assessing the implementation of the test in clinical practice; REFLECTION (NCT05205967) aims to understand the performance of the test in specific clinical settings and its impact on patients and healthcare professionals. Some results from the PATHFINDER study have already been reported. Aiming to evaluate the time and number of additional procedures required to achieve a final diagnosis following a positive test result, it was observed that a cancer signal was detected in 1.5% of participants, of which 65% reached a diagnostic resolution. The median time for diagnosis was 78 days, with 93% of participants undergoing imaging tests and 72% being submitted to an invasive diagnostic procedure. Remarkably, only 18% of participants with a final non-cancer diagnosis had to go through an invasive diagnostic procedure [58,59].

Most PCR- and sequencing-based methods for methylation analysis rely on sodium-bisulfite modification and it has been proven that this chemical treatment causes DNA degradation and fragmentation, hindering the analysis of large CpG islands, especially in cfDNA which is already highly fragmented [60]. As an alternative, immunoprecipitation of methylated DNA (MeDIP), i.e., the use of antibodies that target 5-methylcytosine (5mC) for the enrichment of methylated DNA fragments, followed by sequencing can be used [61]. Following such reasoning, Adela Inc. is developing a sensitive technology for the enrichment of methylated fragments from low input samples, like cfDNA, followed by sequencing of cancer-related regions (cfMeDIP-seq) [62,63]. When applied to cancer detection, by combining the above-described assay with machine learning, AUC values of 0.980, 0.918, 0.971, and 0.969 were depicted for discriminating acute myeloid leukemia, pancreatic cancer, lung cancer, and healthy individuals, respectively. Moreover, early- and late-stage cancer detection depicted similar values [64]. Interestingly, the CAMPERR study (NCT05366881) was, at the time of writing, recruiting patients with any of 20 tumor types, plus healthy individuals to validate the cfMeDIP-seq assay (Table 3).

Several other methylation-based MCED tests using a variety of methodologies are being currently developed by different companies (Table 1). Many of them are also conducting clinical trials for prospective assessment of test performance (Table 3).

Remarkably, methylation analysis showed potential for cancer detection even beyond its molecular analysis. Aberrant DNA methylation patterns in cancer also modify the physicochemical properties of DNA, which led Sina et al. to develop simple, fast analysis and low-input electrochemical and colorimetric assays, achieving AUC values of 0.887 and 0.785 in differentiating breast and colorectal cancer from control plasma samples, respectively [65]. Nonetheless, as only advanced-stage samples were used, although promising, these prototypes require validation in early-stage cancer as well as in more tumor types.

In addition to 5mC, 5-hydroxymethylcytosine (5hmC), another DNA pyrimidine base resulting from 5mC oxidation catalyzed by Ten-Eleven Translocation (TET) enzymes [66], was also proposed as a pan-cancer biomarker by Li et al. [67]. Using genome-wide 5hmC analysis, 67.6% sensitivity and 98.2% specificity were attained for cancer detection and 83.2% accuracy for TOO discrimination in six cancer types [67]. Additionally, BlueStar Genomics is also conducting a study (NCT03869814) for the development of a 5hmC-based MCED test and has already reported some promising preliminary results [68,69].

### 2.3. Fragmentation-Based MCED Tests

The entire population of cfDNA found in the blood of an individual may arise from a wide variety of cell types and its proportions are also dependent on the physiological status. The cfDNA of a healthy individual is primarily derived from dead blood cells, whereas a pathological tissue, such as a tumor tissue, may contribute and release larger amounts of DNA into the circulation [70]. Furthermore, the mechanisms of cell death causing DNA shedding are variable, reflecting different fragmentation patterns, which is also a cell- and tissue-dependent mechanism, reflecting nucleosome positioning in the nucleus [31,70,71]. Thereby, tumor-derived cfDNA fragments carry distinct features that may allow for cancer detection and further TOO identification.

Indeed, this has been confirmed by Bao et al., who showed that combining machine learning algorithms with cfDNA fragmentation profiles enabled lung, colorectal, and liver cancer detection with 95.5% sensitivity and 95% specificity, as well as 93.1% accuracy for TOO prediction, with consistent results even for early-stage and small-size tumors [72]. In this vein, DELFI Diagnostics developed the DELFI assay, which, using genome-wide fragmentation analysis in 236 cancer patients (lung, breast, colorectal, pancreatic, gastric, bile duct, and ovarian) and 245 healthy individuals, displayed 73% sensitivity and 98% specificity for discriminating cancer from healthy subjects, and 61% accuracy for TOO [73,74]. Notably, when combining mutation analysis with fragmentation, DELFI showed an increase in sensitivity to 91% and of TOO accuracy to 75% [74]. Similarly, Mouliere et al. also reported that mutation analysis in size-selected cfDNA fragments detected several cancer types with an AUC over 0.99 [75]. Interestingly, CancerRadar, a multi-omics approach combining cfDNA fragmentation with methylation, copy number variations, and microbial composition depicted a remarkable 85.6% sensitivity and 99% specificity for lung, colon, gastric, and liver cancer detection and 91.5% accuracy for TOO [76].

### 2.4. Gene Expression/Non-Coding RNA-Based MCED Tests

Given their potential as minimally invasive biomarkers for several disorders, the identification of cfRNAs has attracted significant interest in recent years. Circulating microRNAs (miRs) have been the primary focus of cfRNA studies, due to their high abundance and stability in body fluids, as they are often protected by protein complexes and/or within EVs cargo. However, only a small number of miRs exhibit tissue-specificity. Contrarily, messenger RNA (mRNA) and long non-coding RNA (lncRNA) disclose numerous tissue- and disease-specific gene expression patterns and constitute a larger portion of the transcriptome, being easily assessed through RNA sequencing (RNA-seq) [77,78,79].

Supporting the evidence for MCED using whole-transcriptome data, Qi et al. performed RNA-seq in blood samples of 45 cancer patients and 30 healthy individuals and identified 900 differentially expressed genes that were used for constructing a machine learning classifier, which resulted in 0.77 accuracy and 0.72 precision for detecting seven tumor types. Interestingly, when considering only very long intergenic non-coding RNAs (vlincRNAs), the classifier showed 0.86 accuracy and precision, outperforming mRNA-based cancer detection [80]. Other lncRNAs, such as LOC553103 and BLACAT1, also showed the capacity for pan-cancer detection with AUC values ranging from 0.826 to 0.966 and 0.833 to 0.967 for individual cancer types, respectively. Furthermore, discriminating cancer from benign conditions was also achievable in some cases [81,82]. Concerning microRNAs, circulating miR-93 levels were able to detect a variety of different malignancies with 63% to 100% sensitivity and of 81% to 100% specificity for individual cancer types [83]. Moreover, miR-1307-3p also showed 98% sensitivity and 85% specificity in discriminating 13 cancer types from healthy individuals [84]. One advantage of these single-target approaches is that only a simple quantitative PCR reaction is needed, thus favoring clinical implementation.

It has been known for a long time that platelets interact with cancer cells and promote the metastatic cascade at all its phases. Nonetheless, since the interaction between the tumor and the platelets results in the “education” of these particles (tumor-educated platelets, TEPs), altering their transcriptional profile, RNA-seq of TEPs might open a window of new cancer biomarkers [85]. Indeed, Best et al. performed RNA-seq on TEPs from 228 cancer patients and 55 healthy individuals and identified 2246 differentially expressed mRNAs, of which 1072 were selected for constructing a machine learning classifier. This classifier achieved 97% sensitivity, 94% specificity, and 96% accuracy for distinguishing six cancer types from controls, as well as 71% accuracy for TOO prediction [86].

Another interesting approach to multi-cancer detection was reported by Tripathi et al., who developed a scale for scoring individuals as non-cancer, inflammatory, high-risk or stage I–IV cancer. Such a scale was based on OCT-4A expression, a marker of pluripotency, thereby targeting cancer stem cells (CSCs). Remarkably, by enriching CSCs from the blood of 500 cancer patients and 500 non-cancer controls, OCT-4A expression levels detected and staged 22 tumor types with a perfect sensitivity and specificity [87].

### 2.5. Circulating Tumor Cell-Based MCED Tests

CTCs are cells released from the primary tumor into the circulation as a part of the metastatic process. Although usually scarce, the increasing number of CTCs found in the blood has been associated with poor patient prognosis, but its diagnostic and early detection potential remains largely unexplored [88]. Nonetheless, CTCs have been found in the circulation of patients with localized tumors or even prior to the detection of a primary tumor by imaging, thus indicating a putative value in early cancer detection if the right tools are applied [88,89,90,91].

Using EpCAM+/Vimentin+ specific immunomagnetic beads for CTC isolation from 174 cancer patients (118 stage I/II), Huang et al. showed that the mean CTC count in lung, colorectal, gastric, liver, and esophageal cancers was significantly higher when compared to healthy individuals and non-cancer patients with high-risk conditions, also discriminating between the latter two groups of samples [92]. Notably, their technology showed a CTC capture rate higher than 80%, being superior to that of the FDA-approved CellSearch device, which is around 70% [92,93]. Moreover, no significant differences were seen in CTC count between the different cancer types, suggesting a potential multi-cancer detection biomarker, although TOO identification was not possible. Another strategy that has been followed is the analysis of circulating ensembles of tumor-associated cells (C-ETACs), defined as cell clusters with at least 3 cells positive for EpCAM and pan-cytokeratin immunostaining, regardless of CD45 status. C-ETACs detection discriminated cancer patients (18 cancer types) from healthy individuals with 89.8% sensitivity and 97% specificity, outperforming conventional CTCs-based methodologies [94]. Furthermore, the addition of cancer-specific markers to cell staining allowed for TOO identification with 93.1% accuracy [95].

### 2.6. Extracellular Vesicle-Based MCED Tests

EVs are a heterogeneous population of lipid membrane vesicles comprising exosomes, microvesicles and apoptotic bodies, being categorized by size, biogenesis, and release mechanisms particularities [96]. These small vesicles are secreted by a variety of cell types, including cancer cells, and play a major role in mediating cell-cell communication, contributing to the modulation of a cancer-favorable microenvironment. Such a role can be attributed due to EVs transporting different cargo molecules, including nucleic acids, proteins, and metabolites, which are also appealing as cancer biomarkers [97,98].

Unlimited proliferation is a cancer hallmark, due to cancer cells expressing significant levels of telomerase, resulting in the extension of telomeric DNA which hinders cellular senescence [99]. Thus, Goldvaser et al. hypothesized that the presence of human telomerase reverse transcriptase (*hTERT*) mRNA, the catalytical subunit of telomerase, in exosomes could serve as a minimally invasive pan-cancer biomarker. Interestingly, their results sustained the theory, disclosing 62% sensitivity and 100% specificity for detecting 15 cancer types, including solid and hematological ones [100]. Despite the potential of the nucleic acid content of EVs, most studies on MCED have focused on the protein cargo.

Using the Verita™ platform for EV isolation, an alternative to the conventional ultracentrifugation protocols developed by Biological Dynamics, EV-protein profiling combined with machine learning detected stage I and II pancreatic, ovarian, and bladder cancers with 71.2% sensitivity, 99.5% specificity, corresponding to an AUC of 0.95 [101,102]. Focusing on the proteome from both extracellular vesicles and particles (EVPs), machine learning classifiers also discriminated 16 cancer types from healthy controls with 95% sensitivity and 90% specificity using only 47 proteins, while TOO was accurately predicted using a 30-protein classifier [103]. Interestingly, two studies also focused on EVs’ surface proteins and used DNA aptamer-based recognition of such proteins. Using a chip targeting CD9+ EVs and aptamer recognition of CD63/EpCAM/MUC1 (epithelial markers), carcinomas were detected with 100% sensitivity and specificity [104]. When targeting tumor-type-specific proteins in EVs’ surface, 95% sensitivity and 100% specificity were achieved for cancer detection and 68% accuracy for TOO discrimination in cancers of the lung, breast, prostate, liver and ovary, and lymphoma [105].

### 2.7. Other Approaches to MCED Tests

Besides CTCs, other circulating cells have demonstrated the ability to signal several cancer types. Considering that activated monocytes (or macrophages) phagocyte tumor cells or related structures, thereby presenting tumor material highly concentrated in their interior, the epitope detection in monocytes (EDIM) technology was developed [106]. Consisting in the analysis of tumor markers intracellularly of monocytes, this strategy can be easily applied using flow cytometry by targeting CD14+/CD16+ cells extracted from a whole-blood sample in addition to the markers of interest. In fact, this method can be applied to a wide range of diseases, since any epitope may be selected [106].

Combining the natural immune response to cancer with the fact that tumor cells have an altered metabolism, in 2012, Feyen et al. first reported the EDIM-TKTL1 blood test to evaluate if the transketolase-like-1 (TKTL1) protein could be detected in monocytes and allow for cancer detection. This protein was chosen because it is an enzyme involved in the pentose phosphate pathway and upregulated in tumors, thus promoting aerobic glycolysis. Using 240 patients with several malignancies and 117 healthy individuals, the EDIM-TKTL1 test showed 95% sensitivity and 88% specificity, depicting a superior ability to detect small tumors compared to FDG-PET-CT, an imaging technique also relying on cancers’ particular metabolism [107]. Later, Grimm et al. added Apo10, a marker of apoptosis resistance, and showed that the combined analysis of the 2 epitopes (EDIM-TKTL1/Apo10 blood test) detected oral squamous cell carcinoma, breast, and prostate cancer with 95.8% sensitivity and 97.3% specificity [108]. More recently, this technology was further tested on cholangiocellular, pancreatic, and colorectal cancer, showing 100% sensitivity and 96.2% specificity, with the false-positive results being due to individuals harboring inflammatory conditions [109]. In a prospective study involving 5114 asymptomatic individuals, this blood test demonstrated the capacity to be used as a screening tool followed by imaging, since TOO localization is not possible [110]. The EDIM technology has been developed by Zyagnum AG and is available for early cancer early detection as the PanTum Detect^®^ blood test [111].

Pursuing the deregulated metabolism hallmark, metabolite profiling of body fluids may also give insight into the cancer status of individuals. Using different techniques for the analysis of serum metabolites, 84% sensitivity and specificity for detection and 85% accuracy for TOO identification were reported for six cancer types [112], whereas female cancers (breast, endometrial, cervical, and ovarian) could be detected at early stage with 98% sensitivity and 98.3% specificity as well as over 90% accuracy for TOO [113]. Plasma and urine glycosaminoglycans (GAGs) also showed MCED potential, with AUC values around 0.80 [114]. In fact, this GAGome approach to early cancer detection is being tested by Elypta, with two clinical trials currently ongoing (NCT05295017, NCT05235009) (Table 3) [115].

Other components of body fluids, such as metals, also demonstrated capacity for pan-cancer detection, showing an AUC of 0.83 for distinguishing cancer patients from healthy individuals [116]. Remarkably, these elements even disclosed significantly different levels in cancer patients with a normal reading for classical markers (CEA, CA19-9, CA125, PSA). The profile of serum resulting from infrared spectroscopy itself allowed for cancer detection with an AUC of 0.86 [117]. Interestingly, the Dxcover^®^ platform of infrared spectroscopy is being developed as an MCED test, requiring only a blood drop for analysis [118].

When developing cancer biomarkers, it is inevitable not to focus on endogenous molecules, but an exogenous source can also shed light on cancer diagnosis. Although only tested in animal models, non-viral vectors [119] and macrophages [120], engineered to contain a luminescent reporter coupled to the promotor of a tumor-specific actionable gene, showed the ability to point out the presence of very small tumors by measuring luminescence levels in the blood.

**Table 1 cells-12-00935-t001:** Multi-cancer early detection tests validated on human samples.

Biomarker	Source	Tumor Types	Sample	Methods	Main Findings	Test/ Company	Ref.
**DNA** **methylation**	Tissue	lung, breast, colorectal, esophagus, liver, pancreatic, gastric, cervical, head and neck	120 tumor tissue 123 normal tissue	Bisulfite pyrosequencing	TCGA methylation data mining identified *HIST1H4F* as hypermethylated in 17 tumor types. Methylation analysis in tissue samples of 9 cancer types showed AUCs above 0.87 for all cancers and above 0.90 for all except pancreatic cancer.	--	[121]
	Tissue	lung, breast, colorectal, prostate, pancreas, glioblastoma, and B cell chronic lymphocytic leukemia	83 tumor tissue 54 normal tissue	Bisulfite pyrosequencing	Methylation levels at 27 CpGs of the *GHSR* gene showed a higher average methylation degree in all tumor samples compared to normal samples. 27 CpG-signature displayed an AUC of 0.8789 for discriminating cancer from normal tissue.	--	[122]
	Tissue	colorectal, gastric, and esophageal	229 tumor and normal-adjacent tissue	Bisulfite sequencing PCR	TCGA methylation data mining identified differentially methylated regions (DMRs) in the *SST* gene. 7 CpG sites were shown to be hypermethylated in all 3 cancers. A combination of 2 CpGs (+18 and +129) displayed the best AUC of 0.698, with 59.3% sensitivity and 72.8% specificity for detecting the 3 gastrointestinal cancers.	--	[123]
	Tissue	lung, breast, colon, gastric, and endometrial	184 tumor tissue 34 normal tissue	Bisulfite amplicon sequencing	Designed a 302-bp PCR amplicon, covering the *ZNF154* tumor-specific hypermethylated region, and methylation patterns were used to develop a multi-cancer classifier. AUC of 0.96 for discriminating cancer from normal tissue. Computational simulation of ctDNA displayed AUCs of up to 0.79.	--	[124]
	Plasma	colon, pancreatic, liver, and ovarian	71 cancer patients 20 healthy individuals	DREAMing	TCGA methylation data from white blood cells revealed that *ZNF154* locus remains unmethylated, even in older individuals, showing the potential for the development of a blood test for cancer detection. AUC values ranged from 0.75 to 0.87 for discriminating cancer patients from healthy individuals, except for liver cancer which displayed an AUC of 0.48.	--	[42]
	Plasma	lung and prostate	323 cancer patients 136 healthy individuals	qMSP	“PanCancer” panel (*FOXA1*, *RARβ2*, and *RASSF1A*) detected cancer with 64.3% sensitivity, 69.8% specificity and 66.4% accuracy. “CancerType” panel (*GSTP1* and *SOX17*) discriminated between lung and prostate cancer with 93% specificity.	--	[44]
	Plasma	lung, breast, and colorectal	253 cancer patients 103 healthy individuals	qMSP	“PanCancer” panel (*APC*, *FOXA1*, *RASSF1A*) detected cancer with 72.4% sensitivity, 73.5% specificity and 72.8% accuracy. “CancerType” panel (*SCGB3A1*, *SEPT9*, and *SOX17*) discriminated TOO with 80.0%, 98.9%, and 85.1% specificity for breast, colorectal, and lung cancer, respectively.	--	[43]
	Serum	lung, breast, colorectal, gastric, pancreatic, and hepatocellular	70 cancer patients 10 healthy individuals	MSP	Methylation levels of a 4 gene-panel (*RUNX3*, *p16*, *RASSF1A*, and *CDH1*) showed 89% sensitivity and 100% specificity for cancer detection.	--	[125]
	Plasma	colorectal and pancreatic	60 cancer patients 60 healthy individuals	Methylation array	Found a 7 gene panel (*MDR1*, *SRBC*, *VHL*, *MUC2*, *RB1*, *SYK*, and *GPC3*) that detects colorectal and pancreatic cancers with 63.16% sensitivity, 84% specificity, and AUC of 0.8177.	--	[126]
	Plasma	lung, breast, and liver	46 cancer patients 32 healthy individuals	Bisulfite sequencing	Developed CancerLocator, a test based on cfDNA bisulfite sequencing combined with a probabilistic model for cancer detection and TOO discrimination. CancerLocator uses TCGA methylation data as features to estimate the fraction of ctDNA in the plasma and the likelihood of coming from each tumor type. TOO discrimination showed a low error rate of 0.265 (99.7% accuracy).	Cancer Locator	[127]
	Plasma	liver but applicable to any cancer	33 cancer patients 36 healthy individuals	Bisulfite sequencing	Developed CancerDetector, a test based on cfDNA bisulfite sequencing combined with a probabilistic model that joints methylation states of multiple adjacent CpG sites on an individual sequencing read, for cancer detection. 94.8% sensitivity and 100% specificity were obtained.	Cancer Detector	[128]
	Plasma	> 50 cancer types	2482 cancer patients 4207 healthy individuals	Bisulfite sequencing	Developed a targeted methylation assay combined with a machine learning classifier for detecting and discriminating TOO in more than 50 cancer types using cfDNA. 54.9% sensitivity and 99.3% specificity were obtained in the validation set. 93% accuracy for TOO prediction.	Galleri (GRAIL)	[54]
			2823 cancer patients 1254 healthy individuals		Developed a refined assay and classifiers optimized for screening purposes and performed clinical validation. 51.5% sensitivity and 99.5% specificity were obtained. 88.7% accuracy for TOO prediction. PPV of 44.4% and NPV of 99.4% for cancer detection.		[55]
	Plasma	colorectal, hepatocellular, esophageal, gastric, and pancreatic	254 cancer patients 46 healthy individuals	Bisulfite sequencing	Developed EpiPanGI Dx, a cfDNA methylation-based test combining bisulfite sequencing and machine learning, for detecting and discriminating TOO of gastrointestinal cancers. AUC of 0.88 for detecting gastrointestinal cancers. Accuracy of 0.85–0.95 for TOO prediction.	EpiPanGI Dx	[45]
	Plasma	lung, colorectal, gastric, liver, and esophageal	191 pre-diagnosis cancer samples 223 post-diagnosis cancer samples 414 healthy samples	Bisulfite sequencing (using semi-targeted PCR libraries)	Developed PanSeer, a blood test combining the analysis of 477 cancer-specific differentially methylated regions with machine learning for cancer detection. 87.6% sensitivity for post-diagnosis samples, 94.9% sensitivity for pre-diagnosis samples and 96.1% specificity were obtained in the testing set. Cancer can be detected by PanSeer up to 4 years before conventional diagnosis with 95.7% sensitivity.	PanSeer (Singlera Genomics)	[49]
	Plasma	lung, pancreatic, and acute myeloid leukemia	137 cancer patients 62 healthy individuals	cfMeDIP-seq	Developed cfMeDIP-seq, an immunoprecipitation-based protocol for methylation profiling in cfDNA and combined it with machine learning algorithms to discriminate TOO. AUC values ranged from 0.92 to 0.98 for discriminating TOO.	Adela, Inc.	[64]
	Plasma	lung, breast, colorectal, and melanoma	78 cancer patients 66 healthy individuals	Bisulfite sequencing	Developed a targeted methylation sequencing assay to analyze the methylation status of 9 223 cancer related CpG sites, combined with a novel algorithm that converts sequencing data into a methylation score, for cancer detection and TOO discrimination. 83.8% sensitivity and 100% specificity were obtained for cancer detection. 78.9% accuracy for TOO discrimination.	--	[129]
	Plasma	Lung, breast, colorectal, and liver	Not available	NGS	Developed IvyGeneCORE Test, a blood test analyzing cfDNA methylation levels at specific genes combined with artificial intelligence for cancer detection. 84% sensitivity and 90% specificity were obtained for discriminating cancer from healthy individuals.	IvyGeneCORE (Laboratory for Advanced Medicine)	[47]
	Plasma	lung, colorectal, pancreatic, liver, esophageal, and ovarian	625 cancer patients 483 healthy individuals	ELSA-seq	Developed ELSA-seq, a targeted methylation sequencing assay combined with machine learning for cancer detection and TOO discrimination. 80.6% sensitivity and 98.3% specificity were obtained in validation set. 81.0% accuracy for TOO discrimination.	OverC (Burning Rock Dx)	[130]
	Plasma	14 cancer types	549 cancer patients 80 healthy individuals	Targeted sequencing	Developed a cancer detection model based on 37 methylation-correlated blocks (MCB). 72.86% sensitivity, 96.67% specificity, and AUC of 0.86 were obtained in the validation set.	GENECAST	[131]
	Plasma	lung, breast, colorectal, pancreatic, gastric, esophageal, liver, and ovarian	598 cancer patients 302 healthy individuals	Targeted sequencing	Developed a cancer detection and TOO discrimination model based on 135 MCB. 66.3% sensitivity, 95.5% specificity, and AUC of 0.85 were obtained in the validation set. 75.4% accuracy for TOO discrimination.		[132]
	Plasma	lung, breast, colorectal, and pancreatic	101 cancer patients 71 healthy individuals	MSRE-qPCR	Developed a 10-marker panel for cancer detection and a 16-marker panel for TOO discrimination. 79% sensitivity, 90% specificity, and AUC of 0.89 were obtained for cancer detection. TOO discrimination accuracy was 80% for colorectal, 78% for lung, 75% for pancreatic, and 62% for breast cancer.	Signal-X (Universal Dx)	[133]
	Plasma	Lung, colorectal, bladder, and pancreatic	>1500 cancer patients >1800 healthy individuals	5mC enrichment and targeted sequencing	Developed a blood test based on cfDNA methylation signatures for early cancer detection and TOO discrimination. 90% and 87% sensitivity at 90% specificity for stage I/II colorectal and lung cancer detection. 73% and 52% sensitivities at 95% specificity for stage I/II pancreatic and bladder cancer detection. At 98% specificity, TOO accuracy was 99% for colorectal, 94% for lung, 88% for bladder, and 86% for pancreatic cancer.	LUNAR (Guardant Health)	[134]
	Plasma	lung, breast, colorectal, prostate, pancreatic, liver, and ovarian	111 cancer patients 55 healthy individuals	Targeted sequencing	Developed Omni1, a targeted methylation sequencing panel comprising around 3000 cancer-specific hypermethylation markers for cancer early detection. 65% sensitivity for stage I cancers, 75% sensitivity for stage II cancers, and 89% specificity were obtained.	Omni1 (Avida Biomed)	[135]
	Plasma	lung, breast, colorectal, gastric, esophageal, and liver	269 cancer patients 170 healthy individuals	Bisulfite sequencing	Developed Aurora, a blood test based on cancer specific cfDNA methylation signatures for detecting 6 major cancer types. AUCs of 0.90, 0.98, and 0.92 were obtained for lung, breast and colorectal cancer detection, respectively.	Aurora (AnchorDx)	[136]
			203 cancer patients 206 healthy individuals		Improved to Aurora 2.0, a targeted methylation sequencing assay. AUCs of 0.94 and 0.935 were obtained for gastric and esophageal cancer detection, respectively. AUCs of 0.973, 0.962, and 0.92 were obtained for lung, breast, and colorectal cancer detection, respectively.		[137]
			1000 cancer patients 505 healthy individuals		AUCs of 0.973, 0.962 and 0.92, 0.94, and 0.935 were obtained for lung, breast, colorectal, gastric and esophageal cancer detection, respectively. At 99% specificity, 84%, 75%, 82%, 85%, and 78% sensitivity were obtained for lung, breast, colorectal, gastric, and esophageal cancer, respectively.		[138]
	Tissue Plasma	breast, colorectal, prostate, and lymphoma	72 tumor and 31 normal tissues 100 cancer and 45 healthy plasmas	Electrochemical assays	Developed electrochemical and colorimetric assays that can detect methylation differences between cancer and healthy genomes based on the level of DNA adsorption on planar and colloidal gold surfaces. DNA adsorption levels could discriminate between cancer patients and healthy individuals with an AUC of 0.887 using an electrochemical assay. DNA adsorption levels could discriminate between cancer patients and healthy individuals with an AUC of 0.785 using a colorimetric assay.	--	[65]
	Stool	colorectal and gastric	105 cancer patients 113 healthy individuals	Hi-SA	Developed a method combining single-step sodium bisulfite modification and fluorescence PCR to measure *RASSF2* and *SFRP2* methylation status in fecal DNA. DNA recovery from feces showed an AUC of 0.78 for distinguishing cancer from non-advanced lesions (adenomas, polyps and healthy). Methylation levels showed an AUC of 0.78. A combination score showed the best AUC of 0.81.	--	[139]
**DNA** **methylation and** **circulating** **proteins**	Plasma Serum	lung, pancreatic, gastric, esophageal, liver, and ovarian	180 cancer patients 257 healthy individuals	Multiplex PCR and LQAS	Developed a multi-analyte blood test based on 26 methylation markers and 5 circulating proteins combined machine learning algorithms for cancer detection. 83% sensitivity, 94% specificity, and AUC of 0.96 were obtained in the validation set.	Exact Sciences	[140]
			160 cancer patients 315 healthy individuals		85% sensitivity, 95% specificity, and AUC of 0.96 were obtained in the validation set.		[141]
**DNA** **methylation and copy number variations (CNVs)**	Plasma	lung, breast, hepatocellular, nasopharyngeal, smooth muscle sarcoma, and neuroendocrine tumor	46 cancer patients 32 healthy individuals	Bisulfite sequencing	Performed bisulfite sequencing to analyze genome-wide hypomethylation combined with copy number alterations in cfDNA and developed algorithms for cancer detection. If a sample was positive if either hypomethylation or CNAs were observed, 85% sensitivity, and 88% specificity were obtained. If a sample was positive if both hypomethylation and CNAs were observed, 60% sensitivity, and 94% specificity were obtained.	--	[142]
**DNA** **methylation, fragmentation, CNVs and microbial composition**	Plasma	lung, colon, gastric, and liver	275 cancer patients 204 healthy individuals	cfMethyl-Seq	Developed CancerRadar, a test based on genome-wide methylation profiling of cfDNA combined with machine learning for cancer detection and TOO discrimination. 85.6% sensitivity and 99% specificity for cancer detection. 91.5% accuracy for TOO discrimination.	Cancer Radar (Early Diagnostics)	[76]
**DNA** **hydroxymethylation**	Plasma	lung, breast, colorectal, gastric, esophageal, and liver	2241 cancer patients 2289 healthy individuals	5hmC-Seal profiling	Used the 5hmC-Seal technology to profile genome-wide 5hmC in cfDNA and combined it with machine learning for cancer detection and TOO discrimination. 79.3% sensitivity and 95% specificity were obtained in training set. 67.6% sensitivity and 98.2% specificity were obtained in the testing set. 83.2% accuracy for TOO discrimination.	Epican Genetech	[67]
	Plasma	lung, breast, prostate, and pancreatic	188 cancer patients 180 healthy individuals	5hmC sequencing	Developed a novel 5hmC enrichment technology coupled with sequencing and machine learning for cancer detection. AUCs of 0.89, 0.84, 0.95, and 0.83 were obtained for breast, lung, pancreatic and prostate cancer detection, respectively.	BlueStar Genomics	[69]
**Genetic** **variants**	Plasma	lung, breast, colorectal, prostate, bladder, pancreatic, and liver	260 cancer patients 415 healthy individuals	NGS	Developed DEEPGEN^TM^, an assay based on NGS combined with machine learning for cancer detection. 57% sensitivity at 95% specificity, 43% sensitivity at 99% specificity, and AUC of 0.90 were obtained.	DEEPGEN (Quantgene)	[28]
	Stool	colorectal, pancreatic, gastric, biliary, and oropharyngeal	69 cancer patients 69 healthy individuals	Digital melt curve method	Identified target mutations in genes commonly mutated in gastrointestinal cancer by sequencing tumor tissues. Target mutation analysis in stool detected cancer with 68% sensitivity and 100% specificity.	--	[24]
**Genetic** **variants** **and cfDNA fragmentation**	Plasma	lung, breast, colorectal, GIST, ovarian, Hodgkin lymphoma, diffuse large B-cell lymphoma, and multiple myeloma	558 cancer patients 367 healthy individuals	WGS	Developed GIPXplore, a method combining cfDNA whole-genome sequencing profiles with machine learning for cancer detection and TOO discrimination. 92% sensitivity, 98% specificity, and AUC of 0.99 were obtained for discriminating hematological cancers from healthy samples. 85% accuracy for TOO prediction. 55% sensitivity, 95% specificity, and AUC of 0.83 were obtained for discriminating solid cancers from healthy samples. 69% accuracy for TOO prediction.	--	[143]
	Plasma	17 tumor types	200 cancer patients 65 healthy individuals	WGS	Analysis of mutations in size-selected cfDNA fragments improved diagnostic capacity. Combined fragmentation and mutation analysis provided an AUC > 0.99 compared to AUC <0.80 without using fragmentation features.	--	[75]
**cfDNA** **fragmentation**	Plasma	Lung, colorectal, and liver	971 cancer patients 243 healthy individuals	WGS	Used cfDNA fragmentation profiles combined with machine learning for cancer early detection and TOO discrimination. 95.5% sensitivity, 95% specificity, and AUC of 0.983 were obtained. 93.1% accuracy for TOO prediction.	--	[72]
	Plasma	lung, breast, colorectal, pancreatic, gastric, bile duct, and ovarian	236 cancer patients 245 healthy individuals	WGS	Developed DELFI, a test based on cfDNA fragmentation patterns combined with machine learning for cancer detection and TOO discrimination. 73% sensitivity, 98% specificity, and AUC of 0.94 were obtained for discriminating cancer from healthy samples. 61% accuracy for TOO prediction. Combining DELFI with mutant ctDNA, sensitivity for cancer detection increased to 91%, and TOO accuracy increased to 75%.	DELFI (Delfi Diagnostics)	[74]
**Circulating proteins and cfDNA** **mutations**	Plasma	lung, breast, colorectal, pancreas, gastric, liver, esophageal, and ovarian	1005 cancer patients 812 healthy individuals	Targeted sequencing and Bead-based immunoassay	Developed CancerSEEK, a blood test based on cfDNA mutations on 16 genes and 8 circulating proteins combined with machine learning for cancer detection and TOO discrimination. 62% sensitivity, 99% specificity, and AUC of 0.91 were obtained for discriminating cancer from healthy samples. 63% accuracy for TOO prediction.	CancerSEEK (Exact Sciences)	[29]
		--	9911 women not previously known to have cancer		Evaluated the feasibility of CancerSEEK testing combined with PET-CT to detect cancer in a prospective cohort. The blood test was considered positive for 134 participants. 127 were further evaluated by PET. 64 showed imaging concerning for cancer. 26 were proven to have cancer by biopsy or other method. 27.1% sensitivity, 98.9% specificity, and 19.4% PPV were obtained for blood testing alone. 15.6% sensitivity, 99.6% specificity, and 28.3% PPV were obtained for blood testing combined with PET.		[30]
**Circulating Hsp90α**	Plasma	lung, breast, colorectal, stomach, liver, pancreatic, esophageal, and lymphoma	661 cancer patients 308 non-cancer patients 331 healthy individuals	ELISA	Hsp90α levels in plasma discriminated cancer from non-cancer controls (healthy + at-risk). AUC of 0.893, 81.72% sensitivity, and 81.03% specificity were obtained in the validation set.	--	[144]
**Gene** **expression**	Whole blood	lung, breast, colorectal, pancreatic, hepatobiliary, and glioblastoma	228 cancer patients 55 healthy individuals	RNA sequencing	Identified 2246 differentially expressed mRNAs in tumor-educated-platelets (TEPs). 1072 mRNAs were selected for developing a machine learning classifier for multi-cancer detection and TOO prediction. 97% sensitivity, 94% specificity, and 96% accuracy were obtained in the validation set. 71% accuracy for TOO prediction.	thromboDx BV	[86]
	Bone marrow	leukemias, myelodysplastic syndrome, myeloproliferative neoplasm, and lymphoma	136 cancer patients	RNA sequencing	Developed RANKING, a machine learning algorithm applied to RNA-seq data for the identification of hematological cancers. Accuracy of 100% for acute myelocytic leukemia and acute lymphocytic leukemia classification.	--	[145]
	Whole blood	breast, rectum, colon, esophagus, stomach, thyroid, and uterus	45 cancer patients 30 healthy individuals	RNA sequencing	Developed a machine learning classifier that uses RNA-seq data for cancer detection. Identified 900 differentially expressed genes that were used for constructing the classifier. 0.77 accuracy and 0.72 precision were obtained in the testing set. Another classifier based only on very long intergenic non-coding RNAs (vlincRNAs) outperformed the previous with 0.86 accuracy and precision. vlincRNAs demonstrated superior performance compared with mRNAs for cancer status identification.	--	[80]
	Whole blood	22 tumor types	500 cancer patients 500 non-cancer patients	qRT-PCR	Developed the HrC scale, using OCT-4A expression in 120 samples (based on fold increase), for cancer detection and staging. 100% sensitivity, 100% specificity, and AUC of 1 were obtained.	--	[87]
**lncRNA**	Serum	15 tumor types	900 cancer patients 450 patients with benign conditions 450 healthy individuals	qRT-PCR	AUC values ranged from 0.826 to 0.966 for discriminating cancer patients from healthy individuals. AUC values ranged from 0.723 to 0.896 for discriminating cancer from benign conditions. LOC553103 expression was not able to discriminating cancer from benign conditions in pancreatic, ovarian and thyroid cancer.	--	[81]
	Serum	12 tumor types	360 cancer patients 360 patients with benign conditions 360 healthy individuals	qRT-PCR	AUC values ranged from 0.833 to 0.967 for discriminating cancer patients from healthy individuals. AUC values ranged from 0.7 to 0.81 for discriminating cancer from benign conditions. BLACAT1 expression was not able to discriminating cancer from benign conditions in breast, ovarian, prostate, and nasopharyngeal cancer.	--	[82]
**microRNA**	Serum	14 tumor types	112 cancer patients 48 patients with benign conditions 8 healthy individuals	qRT-PCR	Higher miR-93 expression was observed for all cancers compared to healthy controls, except for colorectal, bladder, gastric, renal, cervical, and ovarian cancer. AUC values ranged from 0.86 to 1.00, sensitivities from 63% to 100% and specificities from 81% to 100% for discriminating cancer patients from healthy individuals.	--	[83]
	Serum	13 tumor types	254 cancer patients 27 healthy individuals	microRNA chip	miR-1307-3p levels showed 98% sensitivity, 85% specificity, and AUC of 0.98 for cancer detection in the validation set.	--	[84]
**Epitope detection in monocytes (EDIM)**	Whole blood	17 tumor types	240 cancer patients 117 healthy individuals	Flow cytometry	EDIM-TKTL1 test is based on the detection of activated macrophages that present the TKTL1 antigen intracellularly. 94% sensitivity, 81% specificity and AUC of 0.89 were obtained.	PanTumDetect^®^ (Zyagnum AG)	[107]
		oral squamous cell carcinoma, breast, and prostate	213 cancer patients 74 healthy individuals		Combination of EDIM-TKTL1 and EDIM-Apo10 tests showed 95.8% sensitivity and 97.3% specificity for cancer detection.		[108]
		cholangiocellular, pancreatic and colorectal	62 cancer patients 13 patients with inflammatory conditions 16 healthy individuals		Combination of EDIM-TKTL1 and EDIM-Apo10 tests showed 100% sensitivity, 96.2% specificity and an AUC of 0.9934 for cancer detection. A positive result was seen for 100% of all cancer patients, 0% of healthy individuals, and 7.7% of individuals with inflammation.		[109]
**Circulating ensembles of tumor-associated cells** **(C-ETACs)**	Whole blood	18 cancer types	5509 cancer patients 10,625 healthy individuals	Immuno-staining	C-ETACs were detected in 4944 out of 5509 cancer patients as well as in 255 of the 8493 individuals with no abnormal findings in routine screening procedures. This reflects an 89.8% sensitivity and 97% specificity. C-ETACs were detected in 137 out of 2132 asymptomatic individuals with abnormal findings in routine screening procedures. Assuming that cancer will not clinically manifest in none of the asymptomatic individuals positive for C-ETACs results in a maximum false-positive rate of 3.7%.	--	[94]
	Whole blood	27 cancer types	9416 cancer patients 6725 individuals with suspected cancer 13,919 healthy individuals	Immunocytochemistry	Additional organ-specific markers were profiled aiming to predict TOO. C-ETACs were detected in 91.8% of the 9416 cancer patients. Of the 6725 symptomatic individuals, 6025 were diagnosed with cancer and C-ETACs were detected in 92.6% of these. This resulted in a sensitivity of 92.1%. C-ETACs were undetectable in 13,408 of the 13,919 healthy individuals, resulting in a specificity of 96.3%. 93.1% accuracy for TOO prediction.	--	[95]
**Circulating tumor cells (CTCs)**	Whole blood	lung, colorectal, gastric, liver, and esophageal	174 cancer patients 32 non-cancer patients 25 healthy individuals	Magnetic enrichment and immunofluorescence	CTCs count in cancer patients was significantly higher compared to non-cancer patients with high-risk conditions and healthy individuals (*p* < 0.001). The average CTCs count was 7.3 for cancer patients, 2.4 for non-cancer patients, and 0.9 for healthy individuals. CTCs count superior to 5 could be indicative of cancer status.	--	[92]
**Extracellular** **vesicles (EVs)**	Plasma	breast, lung, acute myelocytic leukemia, and acute lymphocytic leukemia	53 cancer patients 15 healthy individuals	Digital Profiling of Proteins on Individual EV (DPPIE)	Developed an ultrasensitive assay of digital profiling of proteins on individual EV (DPPIE), based on DNA aptamer recognition of specific EV proteins and rolling circle amplification reactions, that produce fluorescent signals on each single EV. DPPIE showed an AUC of 1.0, with specificity and sensitivity of 100% for carcinomas.	--	[104]
	Serum	lung, breast, prostate, liver, ovarian, and lymphoma	145 cancer patients 27 healthy individuals	Thermophoretic aptasensor (TAS)	Developed TAS, an assay based on DNA aptamer recognition of 7 EV proteins and thermophoretic enrichment for cancer detection and TOO discrimination. 95% sensitivity, 100% specificity were obtained for cancer detection and 68% accuracy for TOO discrimination in the validation set.	--	[105]
	Plasma	pancreatic, bladder, and ovarian	139 cancer patients 184 healthy individuals	Verita™ and bead-based immunoassay	Developed an EV-based blood test combining alternating current electrokinetics (Verita™ System) for EVs isolation, immunoassays for protein quantification, and machine learning algorithms for early cancer detection. 71.2% sensitivity, 99.5% specificity, and AUC of 0.95 were obtained for discriminating cancer from healthy samples.	--	[101]
	Plasma	16 cancer types	77 cancer patients 43 healthy individuals	Mass spectrometry	Analysis of tumor-specific extracellular vesicles and particles (EVP) proteomes combined with machine learning allowed cancer detection and TOO discrimination. Based on a 47-protein panel, 95% sensitivity and 90% specificity were obtained in the testing set. Based on all 372 tumor-related proteins, 100% sensitivity and 90% specificity were obtained. 30 protein panel discriminated TOO with very high accuracy.	--	[103]
	Serum	15 cancer types	133 cancer patients 45 healthy individuals	qRT-PCR	Exossomal *hTERT* expression levels detected cancer with 62% sensitivity and 100% specificity.	--	[100]
**Glycosaminoglycans**	Plasma Urine	14 cancer types	753 plasma samples (460 cancers and 293 healthy) 559 urine samples (219 cancers and 340 healthy)	Mass spectrometry	Measured the levels of glycosaminoglycans in plasma and urine samples and combined it with machine learning for cancer detection. AUC of 0.78 was obtained in the validation set for discriminating 5 cancer types from healthy individuals using urine glycosaminoglycans. AUC of 0.84 was obtained in the validation set for discriminating 14 cancer types from healthy individuals using plasma glycosaminoglycans.	GAGome (Elypta)	[114]
**Metabolites**	Serum	lung, colorectal, pancreatic, gastric, liver, and thyroid	950 cancer patients 233 healthy individuals	Mass spectrometry	Developed a laser desorption/ionization mass spectrometry-based liquid biopsy for multi-cancer detection and classification (MNALCI). MNALCI showed 93% sensitivity and 91% specificity for cancer detection in the internal validation cohort and 84% sensitivity and specificity in the external validation cohort. 92% accuracy for TOO discrimination in the internal validation cohort and 85% in the external cohort.	--	[112]
	Serum	breast, endometrial, cervical, and ovarian	1119 cancer patients 250 healthy individuals	Mass spectrometry	Developed a method combining untargeted serum metabolomics with machine learning to identify metabolite signatures that allow early stages cancer detection. 98% sensitivity, 98.3% specificity, and 98% accuracy were obtained. TOO discrimination with 94.4% accuracy for breast, 91.6% for endometrial, 87.6% for cervical, and 92% for ovarian cancer.	--	[113]
**Plasma** **denaturation profiles**	Plasma	glioma but applicable to any cancer	84 cancer patients 63 healthy individuals	Differential scanning fluorimetry	Applied nanoDSF, a differential scanning fluorimetry method for analyzing protein denaturation profiles, to plasma samples and combined it with a machine learning algorithm for distinguish the denaturation profiles of cancer patients. All 5 machine learning algorithms showed accuracies above 87%. Neural Networks (NN) algorithm performed the best, showing 92% sensitivity, 93% specificity and 92.5% accuracy.	--	[146]
**Metallobalance**	Serum	breast, colorectal, prostate, pancreatic, gastric, liver, bile duct, thyroid, ovarian, cervical, and endometrial	1856 cancer patients 5327 healthy individuals	Mass spectrometry	Applied a mass spectrometry-based technology to evaluate the serum profile of 17 elements as a cancer detection tool. AUC of 0.830 for discriminating cancer patients from healthy individuals. Classical markers (CEA, CA19-9, CA125, PSA) alone could not discriminating cancer patients from healthy individuals. However, for individuals with a normal CEA reading, the levels of Na, K, Cu, Fe Co, and Mo displayed differences in cancer over healthy samples, and the same applied to the other classical markers.	--	[116]
**Serum** **spectral** **profile**	Serum	lung, breast, colorectal, prostate, pancreatic, renal, ovarian, and brain	1543 cancer patients 460 symptomatic non-cancer patients 91 healthy individuals	Infrared spectroscopy	Developed Dxcover^®^, an infrared spectroscopy-based blood test for the early detection of cancer and TOO discrimination. 90% sensitivity with 61% specificity (adjusted for higher sensitivity), 56% sensitivity with 91% specificity (adjusted for higher specificity), and AUC of 0.86 were obtained for cancer detection. AUC values ranged from 0.74 to 0.91 for TOO discrimination.	Dxcover^®^	[117]
**Tumor-** **activatable** **minicircles**	Whole blood	Applicable to any cancer	--	Luminescence measurement in the blood	Developed engineered non-viral vectors (minicircles) by coupling SEAP expression to activation of the Survinin promoter, resulting in luminescence production when tumor cells uptake the vectors. Minicircles were injected into tumor-bearing and control mice and SEAP was measured in the blood. AUC of 0.918 was obtained for discriminating cancer from healthy mice.	--	[119]
**Engineered** **macrophages**	Whole blood	Applicable to any cancer	--	Luminescence measurement in the blood	Developed engineered macrophages by coupling luciferase expression to activation of the Arginase-1 promoter, resulting in luminescence production when macrophages adopt an M2 tumor-associated phenotype. Engineered macrophages were injected into tumor-bearing and control mice and luciferase was measured in the blood. 100% sensitivity and specificity were obtained for discriminating cancer from healthy mice.	--	[120]

**Abbreviations:** AUC—Area under ROC curve; cfDNA—cell-free DNA; cfMeDIP-seq—cell-free methylated DNA immunoprecipitation and sequencing; CpG—Cytosine-phosphate-Guanine; ctDNA—circulating tumor DNA; DREAMing—Discrimination of Rare EpiAlleles by Melt; ELISA—enzyme-linked immunosorbent assay; Hi-SA—high-sensitivity assay for bisulfite DNA; LQAS—long probe quantitative amplified signal; MSP—methylation-specific PCR; MSRE-qPCR—methylation-sensitive restriction enzyme -based quantitative PCR; NGS—next generation sequencing; NPV—negative predictive value; PPV—positive predictive value; qMSP—quantitative methylation-specific PCR; qRT-PCR—real-time quantitative reverse transcription PCR; TCGA—The Cancer Genome Atlas; TOO—tissue of origin; WGS—whole genome sequencing; 5hmC—5-hydroxymethylcytosine; 5mC—5-methylcytosine.

## 3. Bioinformatics Meets Cancer Detection: Finding the Right Targets and Improving Biomarker Performance

With the development of The Cancer Genome Atlas (TCGA) project, a large-scale open-access database containing genomic, transcriptomic, epigenomic, and proteomic datasets across more than 30 tumor types, cancer research experienced a boost derived from the analysis of molecular features of individual tumor samples, not only increasing the knowledge on tumor heterogeneity but also on individual and shared profiles among different cancers [147]. Furthermore, given the complexity of sequencing- and array-generated data, many tools have been created to allow a more interactive and comprehensive mining of the different types of data, including the UCSC Cancer Genomics Browser [148], the cBioPortal for Cancer Genomics [149] and UALCAN [150], among others. In 2012, the TCGA Pan-Cancer analysis project was launched with the goal of providing new multi-omics data across multiple cancers, increasing the statistical power of the datasets, and making it easier to identify and analyze common cancer molecular abnormalities [147,151]. Nonetheless, performing differential analysis in the context of early detection and diagnosis also requires a significant amount of data from matched normal tissues. Despite several normal-adjacent tissue datasets being available in TCGA, the sample size is small, which is why many studies combine their analysis with GTEx samples, although only RNA-seq data is available [152]. The Gene Expression Omnibus (GEO) is another database containing many sets of high-throughput sequencing data, consisting in a repository where researchers upload their results and these become freely available to the entire scientific community [153].

The availability of such a large amount of molecular data has prompted data mining as the first step of biomarker discovery and, since the clinical data of the sequenced samples is also publicly available, possibilities range from early detection to prognosis and therapy response prediction. Indeed, many molecules have been proposed as detection biomarkers across several cancer types by data mining (Table 2). For instance, when mining the available methylome data of TCGA in search of pancreatic cancer biomarkers, Manoochehri et al. found DMRs in the first exon of the *SST* gene that were significantly hypermethylated in tissues of 11 cancers compared to para-cancerous tissues [154]. Interestingly, it was later reported that two CpG sites within *SST’s* first exon could detect colorectal, gastric, and esophageal cancer with 59.3% sensitivity and 72.8% specificity using tissue samples of 229 patients [123]. Similarly, the expression of Hsp90α was shown to significantly differ between 9 tumors and respective normal tissues by in silico analysis [155], and the circulating plasma levels of the Hsp90α protein were also reported to detect several cancer types with 81.72% sensitivity and 81.03% specificity [144]. Although without validation in biological specimens, many other markers have shown MCED potential, with the benefit of data mining allowing the reduction of an entire sequencing/array run data into single genes or proteins, enabling the design of a targeted validation assay. For example, Liu et al. used whole-genome methylation data to identify 12 CpG markers and then utilized them to construct a deep learning model that detected 27 cancer types with 92.8% sensitivity, 90.1% specificity, and 92.4% accuracy [156]. Likewise, Ibrahim et al. showed that the methylation levels of a set of 4 CpGs could detect 14 tumor types with an AUC of 0.96 and a set of 20 CpGs discriminated TOO with AUC values ranging from 0.87 to 0.99 [157]. Remarkably, this was possible by using machine learning algorithms that tested different combinations of CpGs to find the most informative ones. Aside from methylation, in silico analysis of the expression of several individual genes, miRs, and lncRNAs also displayed biomarker potential for several tumor types (Table 2), thus being easily validated by quantitative PCR in human samples.

Importantly, these databases also allow the development and training of machine learning algorithms to improve biomarkers’ detection capacity. Fan et al. developed a mathematical model to expand the Illumina 450K methylation array data to cover a larger percentage of the total CpG sites in the genome and combined such expanded data with genome-wide expression and mutational coverage into a random forest classifier that detected cancer with an AUC of 0.85. Additionally, a multi-class regression model was constructed to discriminate TOO in 13 tumors, showing 95.3% accuracy [158]. Applying neural network-based deep learning on transcriptomic data, Yuan et al. developed the DeepDCancer classifier that disclosed 90% accuracy for detecting and an average 94% for discriminating 10 cancer types [159]. Similarly, the GeneCT model was constructed by Sun et al. resulting in 96.0% sensitivity and 96.1% specificity for cancer identification, followed by 99.6% accuracy for TOO prediction [160]. This time focusing on ncRNA, Wang et al. showed that this deep learning approach detected 26 cancers with an AUC of 0.963 and discriminated between cancer types with 82.15% accuracy [161]. MicroRNA data can also be used for classifier construction, with Yuan et al. evaluating several algorithms and reporting accuracies over 95%, but ultimately, the support vector machine (SVM) performed the best with 99% accuracy in classifying 11 tumor types [162].

In fact, the benefits of combining machine learning with molecular analysis may be corroborated by the fact that practically all the studies above mentioned using human samples (Table 1) applied algorithms to the outputted data of the used methodologies and built classifiers for discriminating cancer from healthy and further detecting the underlying cancer type. Indeed, the remarkable results obtained with GRAIL’s MCED test were the result of a custom model based on two ensemble logistic regression (LR) algorithms, one to differentiate cancer from non-cancer and the other to identify the TOO [54]. CancerSEEK’s and PanSeer’s technology also relied on LR [29,49], while DELFI used a stochastic gradient boosting model [74]. This shows that several models exist and can be tested to disclose the most suitable, according to the type of data being used as model features and its final purpose.

Machine learning is a subset of artificial intelligence that uses mathematical and statistical methods to improve a computer’s performance in decision-making. Using large amounts of data, algorithms can be trained to learn certain tasks and then be tested to predict their behavior in a real-world scenario [163,164]. Moreover, deep learning is a subset of machine learning that uses supervised or unsupervised learning methods to train multilayered artificial neural networks. It has been shown to outperform even the best machine learning algorithms, due to performing better on big datasets, which may also be a drawback, since many biological samples are available in low quantity [163,164]. While classical statistics is based on probability and assumption, machine learning uses algorithms that are trained and improved with experience and increasing input data, thus being more effective in dealing with high-resolution data, such as biological data [165]. In fact, it is the complexity of the high-throughput molecular techniques’ data that led to the inclusion of machine learning as a part of biomarker research, in feature extraction of relevant biomarkers, as well as in validating these for sample classification [166]. Moreover, the capacity of developing multimodal algorithms, i.e., models containing not only molecular but also histological, radiological, and clinical data as input features, holds great promise for precision oncology [167]. 

**Table 2 cells-12-00935-t002:** Data mining studies of multi-cancer early detection.

Biomarker	Database	Tumor Types	Main Findings	Ref.
**DNA** **methylation**	TCGA GEO	26 tumor types	Identified 7 informative CpG sites capable of discriminating tumor from normal samples. AUC of 0.986 was obtained in the training set. Validation using GEO datasets of breast, colorectal cancer, and prostate cancer obtained AUCs of 0.97, 0.95, and 0.93, respectively. Validation set comprising the remaining cancer types obtained an AUC of 0.94. Identified 12 CpG sites capable of discriminating each tumor type with an AUC of 0.98.	[168]
	TCGA GEO	27 tumor types	Identified 12 CpG markers and 13 promoter markers and constructed diagnostic models by deep learning. CpG marker model achieved 98.1% sensitivity, 99.5% specificity, and 98.5% accuracy on training set, while achieving 92.8% sensitivity, 90.1% specificity, and 92.4% accuracy on testing set. Promoter marker model achieved 96.9% sensitivity, 99.9% specificity, and 97.8% accuracy on testing set, while achieving 89.8% sensitivity, 81.1% specificity, and 88.3% accuracy on testing set.	[156]
	TCGA GEO	27 tumor types	Developed the CAncer Cell-of-Origin (CACO) methylation panel comprising 2 572 cytosines that are significantly hypermethylated in tumor tissues compared with normal tissues and healthy blood samples. CACO panel identified TOO with AUC ranging from 0.856 to 0.998 in discovery cohort and 0.854 to 0.998 in validation cohort. CACO panel could identify TOO in liquid biopsies and unknown primary carcinoma samples.	[169]
	TCGA	14 tumor types	Combined genome-wide differential methylation profiling with machine learning to detect cancer and discriminate TOO. Set of 4 CpGs detected cancer with an AUC of 0.95 in the discovery set and an AUC of 0.96 in the validation set. Set of 20 CpGs discriminated TOO with AUC values ranging from 0.87 to 0.99; 12 out of 14 cancer types were discriminated with sensitivities and specificities above 90%.	[157]
	TCGA GEO	3 tumor types	Developed a machine learning algorithm to detect and discriminate TOO in 3 urological cancers (prostate, bladder, and kidney) using 128 methylation markers. 99.1% accuracy in training set; 97.6% accuracy in 2 independent validation sets.	[170]
	TCGA GEO	33 tumor types	Identified a 12-market set that can detect all 33 cancers in TCGA database with AUCs > 0.84. Identified sets of 6 markers that can discriminate TOO with AUCs ranging from 0.969 to 1.	[171]
	TCGA	12 tumor types	While performing genome-wide methylation analysis for pancreatic cancer biomarker discovery, identified *SST* as hypermethylated in pancreatic tumors compared to normal tissue and showed an AUC of 0.89 for pancreatic cancer detection in cfDNA. SST methylation and expression in 11 other cancer types showed significant hypermethylation and downregulation of expression when compared to the respective normal tissue (*p* < 0.0001).	[154]
	TCGA GEO	14 tumor types	Identified 6 CpGs in the *GSDME* gene differentially methylated between tumor and normal samples and used them for developing a machine learning algorithm for cancer identification. 98.8% sensitivity, 94.2% specificity, and AUC of 0.86 in the training set. AUC of 0.85 in validation set. 6 CpG model showed TOO discrimination capacity.	[172]
**DNA** **methylation, gene expression and somatic** **mutations**	TCGA	13 tumor types	Developed EAGLING, a model that expands the Illumina 450K array data to cover about 30% of CpGs in the genome. Used this expanded methylation data combined with gene expression and somatic mutation data to identify genes with differential patterns in various cancer types (triple-evidenced genes). Developed a machine learning algorithm, using the identified triple-evidenced genes, for cancer detection. AUC of 0.85 was obtained; 95.3% accuracy was obtained for TOO discrimination. *TNXB, RRM2, CELSR3, SLC16A3, FANCI, MMP9, MMP11, SIK1,* and *TRIM59* showed great capacity for cancer diagnosis.	[158]
**Gene mutations**	TCGA	5 tumor types	Based on a tumor’s mutations and their respective GO terms and KEGG pathways, a machine learning algorithm was developed for TOO discrimination; 62% accuracy was obtained for discriminating TOO in 5 cancer types.	[173]
**Gene expression**	GEO	10 tumor types	Developed a deep learning classifier for multi-cancer diagnosis using transcriptomic data termed DeepDCancer. 90% accuracy was obtained for distinguishing cancer from normal samples, while accuracies ranged from 86 to 98% (94% average) for discriminating individual cancer types. 96% accuracy was obtained for distinguishing cancer from normal samples using an improved classifier, DeepDCancer.	[159]
	TCGA	40 tumor types	Developed SCOPE, a machine learning algorithm that uses RNA-seq data for TOO prediction. SCOPE achieved 97% accuracy in training set and 99% in testing set. SCOPE showed the ability to identify TOO in cancers of unknown primary.	[174]
	TCGA	11 tumor types	Developed GeneCT, a deep learning algorithm that uses RNA-seq data for cancer identification and TOO prediction. Known cancer-related genes were used for cancer status identification and transcription factors for TOO prediction. 100% sensitivity and 99.6% specificity for cancer identification in training set. 96.0% sensitivity and 96.1 specificity for cancer identification in validation set. 99.6% accuracy for TOO prediction in training set and 98.6% in validation set.	[160]
	TCGA	33 tumor types	5 machine learning algorithms were compared on their performance for cancer classification. Linear support vector machine (SVM) showed the best accuracy of 95.8%.	[175]
	TCGA	5 tumor types	Developed a deep learning model for TOO discrimination using RNA-seq data among the 5 most common cancers in women. LASSO feature selection reduced all 14,899 genes to only 173 relevant genes. 99.45% accuracy was obtained for discriminating TOO in 5 cancer types.	[176]
	TCGA GTEx	28 tumor types	Identified differentially expressed genes (DEGs) that were shared in various cancer types and constructed a diagnostic model using 10 upregulated DEGs (*CCNA2, CDK1, CCNB1, CDC20, TOP2A, BUB1B, AURKB, NCAPG, CDC45*, and *TTK*). AUC of 0.894 was obtained for discriminating cancer from normal samples.	[177]
	TCGA	15 tumor types	*MMP11* and *MMP13* expression was significantly higher in most cancer types compared to tissue matched controls. Each cancer type featured at least one MMP with an AUC greater than 0.9, except prostate cancer; 6 cancer types featured 4 or more MMPs with AUC > 0.9. If serum detection is possible, upregulated *MMP11* or *MMP13* could serve as a multi-cancer biomarker.	[178]
	TCGA	9 tumor types	Hsp90α expression was significantly higher in 8 cancers compared to tissue matched controls, except for prostate cancer which displayed significant lower expression. AUC values ranged from 0.63 to 0.94 for individual cancer types.	[155]
	TCGA GTEx	33 tumor types	Claudin-6 was significantly overexpressed in 20 cancer types. AUC > 0.7 were obtained for detecting 15 cancer types. AUC > 0.9 were obtained for detecting acute myeloid leukemia, testicular, ovarian, and uterine cancer.	[179]
	TCGA GTEx	33 tumor types	*YTHDC2* expression was significantly downregulated in most cancers compared with normal tissues. *YTHDC2* displayed high diagnostic value (AUC > 0.90) for 7 cancer types and moderate diagnostic value (AUC > 0.723) in 8 cancer types.	[180]
	TCGA GTEx	24 tumor types	*PAFAH1B* expression was significantly upregulated in most cancers compared with normal tissues. *PAFAH1B* displayed high diagnostic value (AUC > 0.90) for 15 cancer types and moderate diagnostic value (AUC > 0.75) in 9 cancer types.	[181]
	TCGA GTEx	20 tumor types	*SHC1* expression was significantly upregulated in most cancers compared with normal tissues. *SHC1* displayed high diagnostic value (AUC > 0.90) for 4 cancer types and moderate diagnostic value (AUC > 0.70) in 16 cancer types. Strong diagnostic capability for KICH (AUC = 0.92), LIHC (AUC = 0.95), and PAAD (AUC = 0.95).	[182]
	TCGA GTEx	29 tumor types	*GPC2* expression was significantly upregulated in 12 early-stage cancers compared with normal tissues. *GPC2* displayed high diagnostic value (AUC > 0.90) for 6 cancer types, moderate diagnostic value (AUC > 0.70) in 16 cancer types, and low diagnostic value (AUC > 0.50) in 7 cancer types.	[183]
**ncRNA**	TCGA	26 tumor types	Developed algorithms to remove all the factor effects (genetic, epidemiological, and environmental variables) from big data and revealed 56 ncRNAs as universal markers for 26 cancer types. Used these 56 ncRNAs as markers and employed machine learning algorithms to discriminating cancer from normal samples and identify TOO. AUC of 0.963 for discriminating cancer from normal samples. AUC values ranged from 0.99 to 1 for detecting individual cancer types. 82.15% accuracy for discriminating TOO.	[161]
**lncRNA**	TCGA GEO	9 tumor types	CRNDE expression was significantly higher in 9 cancers compared to tissue matched controls. AUC values ranged from 0.855 to 0.984, sensitivities from 70 to 97% and specificities from 75 to 100%. Meta-analysis from 6 studies showed a pooled sensitivity of 77%, specificity of 90%, and AUC of 0.87.	[184]
	TCGA GEO	12 tumor types	Identified 6 differently expressed long intergenic noncoding RNAs (lincRNAs) (PCAN-1 to PCAN-6) and applied machine learning algorithms for cancer detection using 5 of them. AUC of 0.947 was obtained in the training set. AUC of 0.947, 81.7% sensitivity, and 97% specificity were obtained in the testing set.	[185]
	TCGA GEO	8 tumor types	Using RNA-seq and methylation data from TCGA, identified 9 epigenetically regulated lncRNAs (lncRNAs regulated by methylation) that can predict cancer. Developed a score based on expression and methylation data of these 9 genes (PVT1, PSMD5-AS1, FAM83H-AS1, MIR4458HG, HCP5, GAS5, CTD2201E18.3, HCG11, and AC016747.3) that was applied to all cancer and normal samples. AUC values ranged from 0.741 to 0.992 for detecting 8 cancer types. AUC values ranged from 0.712 to 1 in an independent validation set.	[186]
	TCGA	33 tumor types	SNHG3 expression was significantly upregulated in 16 (out of 33) cancers compared with normal tissues. 72% sensitivity, 87% specificity, and an AUC of 0.89 was observed for cancer detection.	[187]
**microRNA**	TCGA	21 tumor types	Used machine learning algorithms to develop a multi-cancer diagnostic method based on microRNA expression. Support vector machine (SVM) classifier was chosen, since it provided the highest accuracy of 97.2%, sensitivities over 90%, and specificities of 100% for most cancers.	[188]
	GEO	11 tumor types	Developed a computational pipeline for extracellular microRNA-based cancer detection and classification. All classifiers showed accuracies over 95%. SVM classifier performed the best, with 99% accuracy. Identified a 10 microRNA-signature capable of TOO discrimination.	[162]
	TCGA	4 tumor types	Identified 3 differentially expressed miRNAs (miR-552, miR-490, and miR-133a-2) with diagnostic potential for digestive tract cancers. 3 miRNAs showed high diagnostic value in rectal cancer (AUC > 0.961) and moderate diagnostic value in esophageal (AUC > 0.826), gastric (AUC > 0.798), and colon cancer (AUC > 0.797).	[189]
	GEO	12 tumor types	Developed a serum-based 4-microRNA diagnostic model (has-miR-5100, has-miR-1343-3hashsa-miR-1290hasnd hsa-miR-4787-3p) for cancer early detection. Sensitivities ranging from 83.2 to 100% for biliary tract, bladder, colorectal, esophageal, gastric, glioma, liver, pancreatic, and prostate cancers were obtained, while reasonable sensitivities of 68.2 and 72.0% for ovarian cancer and sarcoma, respectively, with 99.3% specificity.	[190]
	GEO	12 tumor types	Developed a m6A target miRNAs serum signature, based on 18 microRNAs combined with machine learning, for cancer detection. 93.9% sensitivity, 93.3% specificity, and AUC of 0.979 in training set. 94.2% sensitivity, 91.6% specificity, and AUC of 0.976 in internal validation set. 90.8% sensitivity, 84.7% specificity, and AUC of 0.936 in external validation set.	[191]
**Progenitorness score**	TCGA GEO	17 tumor types	Selected 77 progenitor genes and formulated a score to quantify the progenitorness of a sample using its expression profile data. Tumor samples showed significantly higher progenitorness scores than normal tissues for all cancer types, with AUC ranging from 0.746 to 1.000. For the majority of cancers, AUC was above 0.90.	[192]

## 4. What Is Stopping MCED Tests from Moving into Clinical Application?

MCED tests are paving the way for a shift in the current cancer screening paradigm, moving from single organ screening of a hand full of cancers to universal testing using a simple blood draw. Being a recent topic in cancer research, there are still no guidelines on how to evaluate and compare the performance of tests being developed. For that reason, Braunstein and Ofman proposed 9 criteria that should be taken into account in a MCED test: (1) target high-risk individuals; (2) detect the highest possible number of cancer types; (3) display a low false-positive rate (FPR); (4) possess an accurate TOO discriminating capacity; (5) limit detection to cancers that tend to be deadly during a typical lifetime; (6) display a balanced sensitivity and specificity; (7) be validated in prospective, multi-center, population based studies; (8) be evaluated in studies comprising several controls, namely, non-cancer individuals and those harboring benign and inflammatory conditions; (9) be cost-effective [193]. Still, even if such criteria are met, the performance of MCED tests in a real-life screening scenario remains unknown.

Thus far, MCED studies have only reported tests’ diagnostic capacity by using post-diagnosis samples in a retrospective case-control manner [194,195]. Hence, it is not possible to know if these tests will indeed detect cancer at early stages and, if so, if such stage-shift will result in clinical benefit and mortality reduction. Furthermore, there are no guidelines for confirming a positive test result and, consequently, there is no way to infer the real false-result rates and TOO accuracy, as well as the number of necessary additional procedures to reach a final diagnosis. Importantly, questions regarding when, how, and to whom to provide a MCED test also remain unanswered [194,195,196]. To obtain insights into such uncertainties, prospective randomized clinical trials, comparing screened and unscreened asymptomatic individuals, with the right endpoint being cancer-specific mortality, are demanded [194,195]. Nonetheless, these studies require many participants and a long follow-up to provide meaningful clinical information. Interestingly, Hackshaw and Berg suggested that the adoption of a nested randomized trial i.e., storing collected blood samples and only applying the test in those positive, whether in the screened group or in the control group, would allow for a more economic and less resource-consuming trial [197]. Currently, most of the developed MCED tests are under prospective trials, either in asymptomatic general or high-risk populations, to confirm their screening potential and shed light into all the above-mentioned concerns (Table 3).

Notably, many simulation studies have used the available performance data of published MCED tests to estimate their potential impact on health care systems. For instance, combining stage-specific incidence and survival data from SEER with GRAIL’s test performance, a 78% reduction in late-stage cancer incidence and 26% of all cancer-related deaths were estimated [198]. In addition, breast cancer detection might be incremented in 11% if a MCED test would be available during a routine examination [199]. When estimating the impact of pan-cancer screening in the US and UK, Hackshaw et al. reported that using such a strategy in someone without a cancer detected by conventional screening not only increased cancer detection rates, but also significantly reduced the cost of additional diagnostic workups [200]. Similarly, Tafazzoli et al. demonstrated an estimated USD 5421 in cost reduction per cancer treatment if, hypothetically, annual MCED testing was provided to the population between 50 and 79 years [201]. Focusing on the consequent harms and benefits, a favorable balance was depicted between the number of individuals exposed to unnecessary confirmatory tests and the number of detected cancers, with it being even improved if the test included more lethal cancers [202]. In fact, the harms of MCED testing, not only concerning overdiagnosis and needless procedures, but also the resulting anxiety and the overall perception of screening, should not be overlooked. To be cost-effective, population adherence to screening programs must be high, thus psychological and behavioral aspects also need to be taken into account, such as how the science behind the tests will be explained, the tests be delivered, and the results revealed, or further procedures be recommended [203]. Hence, for a successful implementation of MCED tests, adequate medical communication and public understanding is required [203].

Another concern regarding MCED tests is the use of high-throughput sequencing-based methodologies. Although allowing for the simultaneous screen of several genomic regions, which is beneficial when using a limiting material as liquid biopsies, such methods are highly costly and lengthy processes, requiring specialized data analysis [204,205]. As mentioned above, tests may reach USD 1000 per individual, which is not a feasible option for population-based screening. Thus, efforts should also be made to develop more targeted, fast workflow and cost-effective assays.

Notably, lack of standardization is a major issue encompassing the liquid biopsy research field. Pre-analytical variables and isolation methods have great impact on subsequent molecular results, precluding an accurate comparison between different tests and studies. Hence, initiatives such as Cancer-ID, now replaced by the European Liquid Biopsy Society (ELBS), and BloodPAC are key for standardizing and moving liquid biopsy testing from bench to beside [206,207].

**Table 3 cells-12-00935-t003:** Clinical trials conducted/ongoing for validation of multi-cancer early detection tests.

Trial ID	Trial Name	MCED Test	Sponsor	Status (as of July 2022)
NCT02889978	CCGA	Galleri	GRAIL, LLC	Active, not recruiting
NCT03085888	STRIVE	Galleri	GRAIL, LLC	Active, not recruiting
NCT03934866	SUMMIT	Galleri	University College London and GRAIL	Active, not recruiting
NCT04241796	PATHFINDER	Galleri	GRAIL, LLC	Active, not recruiting
NCT05155605	PATHFINDER 2	Galleri	GRAIL, LLC	Recruiting
NCT05205967	REFLECTION	Galleri	GRAIL, LLC	Recruiting
NCT05235009	LEV87A	GAGome	Elypta	Recruiting
NCT05295017	LEV93A	GAGome	Elypta	Recruiting
NCT05227534	PREVENT	OverC	Guangzhou Burning Rock Dx	Recruiting
NCT04825834	DELFI-L101	DELFI	Delfi Diagnostics Inc.	Recruiting
NCT04213326	ASCEND	CancerSEEK	Exact Sciences	Completed
NCT03756597	PAN	ReCIVA	Owlstone Ltd.	Unknown
NCT03517332	--	DEEPGEN	Quantgene Inc.	Unknown
NCT03967652	--	Na-nose	Anhui Medical University	Not yet recruiting
NCT05366881	CAMPERR	--	Adela, Inc.	Recruiting
NCT05254834	Vallania	--	Freenome Holdings Inc.	Recruiting
NCT04972201	PROMISE	--	Chinese Academy of Medical Sciences (and Burning Rock Dx)	Recruiting
NCT04822792	PRESCIENT	--	Chinese Academy of Medical Sciences (and Burning Rock Dx)	Recruiting
NCT04820868	THUNDER	--	Shanghai Zhongshan Hospital (and Burning Rock Dx)	Recruiting
NCT04817306	PREDICT	--	Shanghai Zhongshan Hospital (and Burning Rock Dx)	Recruiting
NCT05227261	K-DETEK	--	Gene Solutions	Recruiting
NCT05159544	FuSion	--	Singlera Genomics Inc.	Recruiting
NCT04405557	PREDICT	--	Geneplus-Beijing Co.	Active, not recruiting
NCT02662621	EXODIAG	--	Centre Georges Francois Leclerc	Completed
NCT04197414	--	--	Yonsei University	Recruiting
NCT03951428	--	--	LifeStory Health Inc.	Unknown
NCT03869814	--	--	Bluestar Genomics Inc.	Active, not recruiting
NCT02612350	--	--	Pathway Genomics	Completed

## 5. Conclusions

Overall, by complementing the currently available screening and diagnostic approaches, MCED tests show great promise for reducing cancer mortality by shifting detection to earlier stages, in which curative options are most likely to be effective. DNA methylation-based tests are in forefront of development, being the most frequently chosen source of biomarkers, due to its features of aberrant tumor-specific patterns, tissue-specificity, and easiness to assess in cfDNA. By combining molecular analysis of liquid biopsies with artificial intelligence, the performance of MCED tests may be greatly improved, increasing not only the sensitivity of detection of multiple cancers but also the accuracy of discriminating among the different tumor types (Figure 3). MCED tests, however, still lack validation in prospective multicenter studies to enable their implementation into population-based screening programs and make their way into routine clinical practice.

## Figures and Tables

**Figure 1 cells-12-00935-f001:**
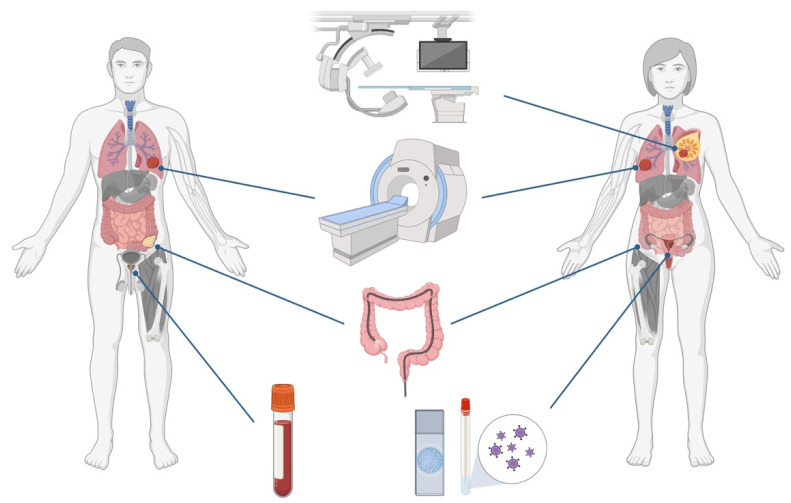
Currently available cancer screening options: mammography for breast cancer; low-dose CT for lung cancer; colonoscopy for colorectal cancer; cytology and HPV testing for cervical cancer; serum PSA testing for prostate cancer. Colored organs represent those with available screening; grey organs represent those without any current screening option (not all cancer types are represented). Created with Biorender.com.

**Figure 2 cells-12-00935-f002:**
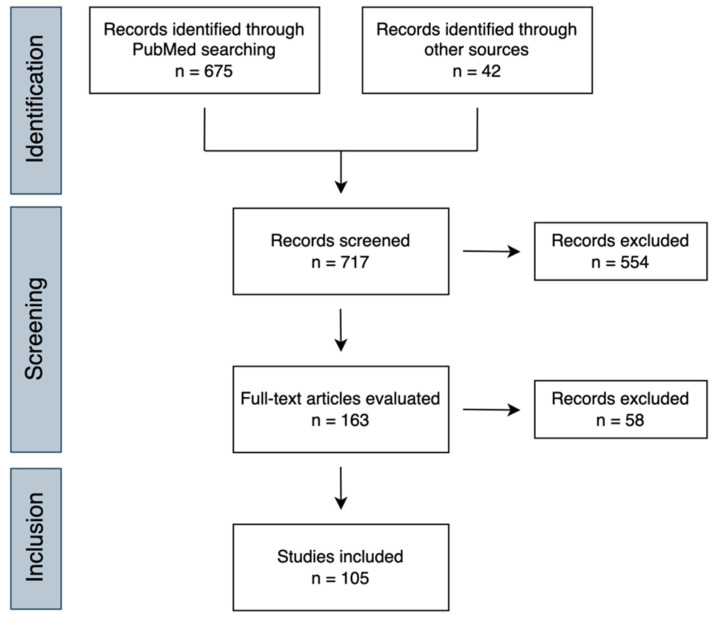
Flow diagram of the conducted search methodology for this review.

**Figure 3 cells-12-00935-f003:**
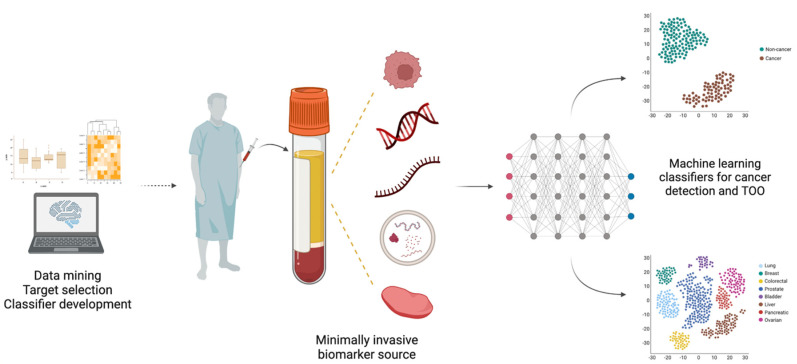
Schematic representation of the workflow for developing multi-cancer early detection tests. Data mining of big datasets, such as TCGA, are great tools for selecting biomarkers with utility for cancer detection. Liquid biopsies provide a minimally invasive way to obtain cancer-related information, namely, circulating tumor cells (CTCs), circulating nucleic acids (cfDNA and cfRNA), extracellular vesicles (EVs), and tumor-educated platelets (TEPs). The molecular analysis of these biomarkers combined with machine learning classifiers shows great potential for detecting multiple cancers simultaneously and discriminating tissue of origin (TOO). Created with BioRender.com.

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
