# Peer review of "Shifting the Cancer Screening Paradigm: The Rising Potential of Blood-Based Multi-Cancer Early Detection Tests"

_cells, 2023, doi:10.3390/cells12060935_

Round 1
Reviewer 1 Report
This is a very interesting review evaluating tha state of art of reserach on minimally invasive tools for early cancer detection (MCED) tests which combine molecular analyses of tumor-related markers present in body fluids with artificial intelligence to simultaneously detect and differentiate cancer types. The review provides a highlight of the different strategies under development and discuss the major factors which are preventing clinical implementation.
I find the review overall complete and well written addressing all the issues on the topic.
Author Response
Dear reviewer,
thank you for the positive comments on our manuscript.
Reviewer 2 Report
the paper is very well written, with a huge search of data. It is a very useful review. I wrote some comments about literature data in the text ( in comments).

Author Response
Dear Reviewer,
Thanks for the constructive and positive feedback. Based on the comments, we have now made the requested alterations in the manuscript and marked them with track changes. In this cover letter we include a point-by-point response to the comments of the reviewers.
We believe that the manuscript is improved after adding the recommended edits. We would be grateful if you could consider our revised article for publication.
Response to Reviewer 2:
The paper is very well written, with a huge search of data. It is a very useful review. I wrote some comments about literature data in the text (in comments).
R: The authors thank the reviewer for the positive feedback and for the comments. Here we will give a brief answer to the comments made. All alterations are now included in the newly submitted manuscript.
- I am not sure if ctDNA was the first compartment of liquid biopsy to be used in clinical practice. I heard that MD Anderson used in 2010´s CTCs to follow-up metastatic prostate cancer, together with PSA levels. Please, check this information.
R: The authors thank the reviewer for the relevant comment. Indeed, the first used and also FDA approved liquid biopsy assay was based on CTCs, not on ctDNA. It is now corrected on the manuscript. In section 2.1, lines 115-117, we were referring to ctDNA as the most currently used and useful analyte in clinical practice.
- There are 2 important papers missing: Illie et al., 2014 and Crock et al., 2022. I know that these two papers are not for muti-tumor detection, but they show that CTCs can be used for early detection, although, alone, not able to point where they come from.
R: The authors are thankful for the suggested articles and now included them in the manuscript.
- It seems that Cancer-ID had finished but did not publish results yet. Please, check it.
R: The authors thank the reviewer for the attentive comment. Indeed, Cancer-ID was a five-year consortium that came to an end in 2019, having published best-practice protocols and the results of studies based on the implementation of such protocols. However, a novel EU consortium called the European Liquid Biopsy Society (ELBS) was established and replaced Cancer-ID, thereby continuing to pursue its goals. A mention to this new consortium is now included in the manuscript.